# Level of and factors associated with optimal uptake of intermittent preventive treatment for malaria in pregnancy at private-not-for-profit health facilities in Kasese district

Julius Mutoro[1]*, Justus Barageine Kafunjo[2], Freddy Eric-Kitutu[3], Joan Kalyango[1], Iving Mumbere[1], Nathan Nshakira[4]

1 Clinical Epidemiology Unit, College of Health Sciences, Makerere University, Kampala, Uganda,
2 Department of Obstetrics and Gynaecology, College of Health Sciences, Makerere University, Kampala, Uganda, 3 Department of Pharmacy, School of Health Sciences, Makerere University, Kampala, Uganda,
4 Department of Community Health, School of Medicine, Kabale University, Kampala, Uganda

* juliusmutoro@gmail.com

**Data Availability Statement:** All data are in the manuscript and/or supporting information file.

## Abstract

Malaria in pregnancy poses a high risk of poor maternal and neonatal outcomes and WHO recommends IPTp. However, its uptake has remained sub-optimal among mothers who attend antenatal care at private-not-for-profit health facilities. This study determined the level of and factors associated with uptake Intermittent preventive treatment of malaria for pregnant women (IPTp) at private-not-for-profit (PNFP) health facilities in Kasese District, Uganda. This was a cross-sectional study involving 396 postpartum mothers in the postnatal wards of 8 PNFP health facilities in Kasese district was conducted in September 2022. One hospital and 2 Health Centre IVs were purposively selected and 5 Health Centre IIIs selected randomly. Mothers were consecutively selected and interviewer administered semi-structured questionnaires were used to collect the data. Data were entered in epi-data version 3.1, cleaned and analyzed using STATA version 14. Data were adjusted for clustering & modified poison regression was used to determine associations of the factors and the outcome. From the analysis, level of optimal uptake of IPTp was 51.5% CI = (46.6–56.4). Being married (aPR = 1.35, 95% CI = 1.06–1.7, p = 0.014), attending ANC more than 4 Visits (aPR = 1.29, 95%CI = 1.09–1.54, p<0.001) positively influence optimal uptake while not taking IPTp at recommended time intervals (aPR = 0.49, 95%CI = 0.39–0.62 p<0.001) and mothers paying for IPTp drugs themselves (aPR = 0.74, 95%CI = 0.57–0.97, p = 0.031) negatively influence optimal uptake. This moderate uptake of IPTp among pregnant mothers suggests insufficient protection of pregnant mothers against malaria. Efforts to improve Antenatal care attendance, taking IPTp at recommended time intervals, not paying for ITPp drugs and encouraging marriages should be intensified.

**Funding:** The authors received no specific funding for this work.

**Competing interests:** The authors have declared that no competing interests exist.

# Background

Globally Malaria is a major public health problem with an estimated 219 million cases and 435,000 deaths [1]. Sub-Saharan Africa continuously has the highest incidence of malaria among the risky populations translating to 95% of the global burden of malaria cases [2]. In Africa, malaria in pregnancy poses a 3–4 times risk of miscarriage, low birth weight and increased stillbirth [3]. In Uganda, malaria remains the highest contributor (29.1%) of outpatient cases and with 10.9% of all deaths [4]. Additionally, the five malaria reference centers across Uganda have further recorded 69.7% malaria positivity rate at their outpatient departments [5]. In high malaria transmission settings in Uganda, the risk of placental malaria accounts for 44.6% among pregnant mothers [6]. In response, Uganda adopted the WHO recommendations which call for daily co-trimoxazole for pregnant women who are HIV positive and atleast 3 doses of IPTp-SP throughout for pregnant women who are HIV negative, a schedule at one dose a month, beginning from second trimester of pregnancy [7]. Although an analytical cross sectional study linked optimal uptake of IPTp to reduction of malaria infections by 73.3% and anemia by more than half [8]. Optimal uptake of IPTp is still very low (14.7%) across Uganda [9] due to multiple challenges that include: inadequacies in knowledge of pregnant women on IPTp, side effects of the drugs and stock outs of IPTp supplies that limit optimal uptake in Uganda [10]. Though policies and guidelines are in place on IPTp, there is still a challenge in ensuring adherence to optimal uptake of IPTp and this is evidenced by the insufficient monitoring mechanism of ensuring that mothers take the drug while away from the facility in the event that they did not take it as Directly Observed Treatment (DOT). In Kasese District, Private-not-for-profit (PNFP) health facilities are located in mountainous and densely populated areas and they constitute 11% (15/128) of the health facilities and offer Antenatal care (ANC) and maternity services to about 40% of the mothers in the communities. These PNFPs charge user fees for health services which they also extend to essential services of antenatal care inform of paying for antenatal cards and some investigations like Urinalysis, Hemoglobin tests etc. and this is likely to limit access of women with low incomes [11]. Numerous research on IPTp uptake have been conducted in public facilities, but they have not focused on pregnant women who use PNFP health facilities, which are the only closest facilities available to them in their hard–to-reach areas. If the situation persists, mothers may be more susceptible to malaria complications during pregnancy and the subsequent impacts on unborn babies. This justifies the need to determine the level and factors associated with optimal uptake of intermittent preventive treatment (IPT) for malaria during pregnancy in private, non-profit health facilities in the Kasese district of Uganda.

The Ministry of Health will utilize the data to strengthen implementation of the policy and guidelines on the prevention of malaria. At facility level, it will enable health workers to identify important factors affecting IPTp and devise facility-based plans for improving the situation.

# Methods

## Study setting

This study was conducted at postnatal wards of 8 PNFP health facilities (1 hospital, 2 HCIV, 5 HCIIIs) in Kasese District which is predominantly rural and mountainous located approximately 350Km west of Kampala. The District borders with the Democratic Republic Congo (DRC) in the west and has 128 health facilities (4 hospitals, 5 Health centre (HC) IVs, 45 HC III, 74 HC II), among which 15 are PNFPs (1 hospital, 2 HC IV, 11 HCIII & 1 HCII). At the health facilities, the study was conducted in the postnatal wards of the selected health facilities

because we expected mothers in postnatal ward to have completed the antenatal cycle. All the selected health facilities offer antenatal and postnatal care services and they have different working conditions and environments and these provide differential contexts for providing IPTp.

## Study design, participant enrollment, and data collection

The study was a cross sectional study that included mothers in post-natal ward having delivered within 72 Hours, had attended ANC from any PNFP facility in the district and gave informed consent. Those who had delivered from other health facilities but were referred to the study site for special care of the infant or the mother were excluded from the study. Purposive sampling for 3(1 hospitals & 2HC IVs) and random sampling for 5 HCIIIs from 11 HC IIIs using the list of facilities from DHO's office was done. These facilities included: Kagando Hospital, St. Puals HCIV, Rwesande HCIV, Katadoba HCIII, Kasanga PHC HCIII, Kyarumba PHC HCIII, Nyabugando HCIII & Kinyamaseka HCIII. In this study, Hospitals and HCIVs were purposively because they were the only highest level PNFP facilities in the district with large numbers of post-partum mothers and located in different Health sub districts. HC IIs were not selected because they do not offer delivery services. Proportionate sampling was used to select the number of participants per health facility and consecutive sampling done for participants. The number to be sampled from each facility was calculated by dividing the population size of that health facility by the total population of all facilities and multiplying the resultant proportion by the sample size and this is shown in Table 1.

Data were collected from 396 participants using semi-structured questionnaires & verified using ANC cards by 8 trained research Assistants. These the questionnaires were translated into Lhukonzo as the commonly spoken language.

## Sample size and power calculations

The sample size for the study was calculated using the formula for single proportions (modified Kish Leslie) [12] to cater for design effect. $N = (Z^2\alpha/2\rho(1-\rho)/d^2)$ *DE. Where: N = 396, Z is the standard normal value corresponding to 95% level of confidence which is 1.96, P is the percentage of mothers who took 3 or more doses of IPTp-SP, 14.7%, [9] q is 1-p, = 1–0.147, d is the tolerable sampling error (precision) estimated at 5%, DE- is design effect, We multiplied by 2 to cater for the design effect. The minimum required sample size estimated was 386, However, we had collected extra 10 questionnaires and they were also added to make 396.

**Table 1. Shows number of participants sampled per health facility.**

| Health facility | Deliveries for August 2022 | Number Interviewed |
|---|---:|---:|
| St. Pauls HCIV | 200 | 120 |
| Kagando Hospital | 165 | 103 |
| Rwesande HCIV | 65 | 50 |
| Nyabugando HCIII | 56 | 30 |
| Katadoba HCIII | 78 | 25 |
| Kyarumba PHC HCIII | 80 | 21 |
| Kasanga PHC HCIII | 62 | 22 |
| Kinyamaseka HCIII | 42 | 25 |
| **Total** | **748** | **396** |

## Statistical analysis

Data were transferred from the excel spread sheets into STATA version 14.0 for analysis and missing data were checked for each variable. Data was adjusted for clustering. To calculate the proportion of mothers who received optimal IPTp, we divided the number of participants who took three or more doses of IPTp-SP plus number of HIV positive mothers with adherence of >95% to contrimoxazole by total number of eligible study participants. At univariate level, the data was presented as percentages and frequencies and categorical variables, medians with corresponding interquartile ranges and means and standard deviation for continuous variables. At Bivariate analysis, the association between the outcome (optimal uptake of IPTp) and the independent variables was done using modified Poisson regression reporting prevalence ratios (PRs). A bivariate model was run with the dependent variable (Uptake of three or more doses) with each of the independent variable each at a time to obtain un-adjusted prevalence ratios between the outcome and each predictor. Independent variables with p-values less than 0.2 were selected for multivariate analysis. The selected variables from bivariate analysis (p<0.2) were then run in a multivariable model and then stepwise backward removal was performed to determine which factors remained significant at p<0.05. At this stage only 4 Variables remained significant in the model and then interaction terms were formed between them. A stepwise backward elimination method was then applied removing variables with the largest non-significant p values (p>0.05) systematically until only significant variables and those that improved the fit of the model were retained. The significant variables from the stepwise regression formed the crude model for the multivariable analysis to which confounding was assessed. A variable was considered to be a confounder when it changed the prevalence ratio by 10% or more and would be included in the final multivariable model. The prevalence ratios (PR) and 95% confidence intervals were presented as the measure of association between the different predictors and uptake of three or more doses.

## Ethical considerations

The data collection for this cross sectional study was conducted between 8[th] sept and 23[rd] September 2022. Ethical approval was also obtained from the School of Medicine Research Ethics Committee (SOMREC) with the study number Mak-SOMREC-2022-398. Permission was also obtained from Clinical Epidemiology Unit (CEU), Kasese District Health Office and the respective health facilities where the study was conducted. Written informed consent were obtained from participants. Mothers below age of 18 years consented on assumption that they were emancipated minors. Though the participants consented to the study, they still had a right to discontinue their participation at any time of the study.

# Results

## Baseline characteristics

A total of 396 postpartum mothers from eight health facilities in Kasese district participated in the study between 8[th] sept and 23[rd] September 2022. During data collection, 10 more questionnaires to make 396 were collected and they were included in the analysis since we did not want to lose information. Study participants had a mean (+/- SD) age of 26.1 (6.8) years and 269 (67.9%) were married, about half 200 (50.5%) had primary as highest level of education and 272(68.7%) and resided in rural areas, 270 (68.2%) were unemployed, 230(58.1%) live in a distance of less than 5 Km from the health facility and 305 (77.0%) lived with their husbands. Details of socio-demographics of the participants are shown in Table 2.

**Table 2. Demographic characteristics of 396 mothers from 8 Health facilities in Kasese.**

| Characteristic | Category | n(%) |
|---|---|---|
| Age | ≤26 years | 212(53.5) |
| | >26 years | 184(46.5) |
| Marital status | Single | 101(25.5) |
| | Married | **269(67.9)** |
| | Divorce | 26(6.6) |
| Education level | No education | 66(16.7) |
| | Primary | **200(50.5)** |
| | Secondary | 105(26.5) |
| | Tertiary | 25(6.3) |
| Residence | Rural | **272(68.7)** |
| | Urban | 124(31.3) |
| Occupation of participants | Employed | 32(8.1) |
| | Self employed | 94(23.7) |
| | Unemployed | **270(68.2)** |
| Distance | Less than 5km | **230(58.1)** |
| | More than 5km | 166(41.9) |
| Mothers living with their husbands | Yes | **231(58.3) (77.1)** |
| | No | 38(9.6) |
| | Not applicable | 127(32.1) |

n-Number of participants, %-Percentage

## Health seeking behaviors, IPTp drug factors and knowledge characteristics of respondents

From the analysis, 211(53.3%) of the mothers attended ≥4 antenatal care visits and 250 (63.1%) mothers were escorted by their husbands. Also majority, 281 (71.0%) mothers paid less than $2.67 at each ANC visit. Additionally, 203(57.2) took IPTp at recommended time intervals with 56(15.8) of the mothers who met the costs for IPTp drugs at the health facilities they visited for ANC. The median parity for mothers was 2 and it ranged from 2 to 4. More than three quarters of the mothers 306(77.3%) were multiparous and a third had a gestation age of 1–14 weeks 290(73.3%) at first ANC and 300(84.5%) had inadequate knowledge on IPTp as shown in Table 3 below

## Level of optimal uptake of three or more doses of IPTp

From the 396 participants, 204 mothers took ≥3doses of IPTp-SP or ≥95% adherence to cotrimoxazole giving an overall prevalence of 51.5% (46.6–56.4). The prevalence varied across health facilities with the highest prevalence seen at Kagando Hospital (61.2%) and the lowest at Kyarumba PHC HCIII (33.3%)

## Factors associated with uptake of three or more doses of IPTp

At Bivariate analysis, 10 out of 20 variables were associated with optimal uptake of IPTp at p<0.2. they included: Marital status (CPR. = 1.51 95%CI 1.15–1.99), occupation (CPR. = 0.73 95%CI 0.53–0.99), and Distance from health facility (CPR. = 084, 95%CI 0.0.68–1.02), number of ANC visits ≥4 (CPR. = 1.45 95%CI 1.19–1.56), mothers escorted by husbands (CPR. = 0.86, 95% CI 0.71–1.05), taking IPT as Directly observed therapy(DOT) (CPR. = 0.69 95%CI 0.57–

**Table 3. Health seeking behaviors, IPTp drug and knowledge characteristics of mothers from 8 PNFP health facilities.**

| Characteristic | Category | N(%) |
|---|---|---|
| Number of ANC visits for mothers (N=396) | Less than 4 | **211(53.3)** |
| | 4 or more 4 | 185(46.7) |
| Mothers escorted by their husbands | Yes | 146(36.9) |
| | No | **250(63.1)** |
| Amount paid per ANC visit (N=396) | <$2.6 | **281(71.0)** |
| | $2.96-$5.34 | 69(17.4) |
| | >$5.34 | 46(11.6) |
| Taking IPTp at recommended time intervals (N=355) | Yes | **203(57.2)** |
| | No | 152(42.8) |
| Who paid costs for IPTp drugs(N=355) | Did not buy drugs | **249(70.1)** |
| | Self | 56(15.8) |
| | husband/parent | 50(14.1) |
| Parity(N=396) | Median(IQR) | 2(2-4) |
| Gravidity(N=396) | Prime-parous | 90(22.7) |
| | Multi-parous | **306(77.3)** |
| Gestational age at ANC visit –weeks (N=396) | 1-14 | 79(19.9) |
| | 14-27 | **290(73.3)** |
| | 28-40 | 27(6.8) |
| Knowledge of mothers on Malaria in pregnancy(N=396) | Not adequate | **281(70.9)** |
| | Adequate | 115(29.1) |
| HIV status | HIV Positive | 31(7.8) |
| | HIV negative | 365(92.2) |
| Type of IPTp Medicine taken | SP | **300(84.5)** |
| | Cotrimaxazole | 20(5.6) |
| | Don't know | 35(9.9) |

N-Number of participants, %-Percentage

094), Taking IPTp at recommended time intervals (CPR. = 0.49, 95%CI 0.38–0.62), malaria status of mothers in the last three months (CPR. = 1.26 95%CI 0.99–0.1.61), mothers knowledge (CPR. = 1.14 95%CI 0.93–1.39) as shown in Tables 2 and 3.

At multivariate analysis: four variables were found statistically significant at P<0.05 and they include: being married, (aPR 1.35, 95%CI 1.06–1.7, p = 0.014), attending ANC more than 4 Visits, (aPR = 1.29, 95%CI = 1.09–1.54, p = <0.001), IPTp drugs not taken at recommended time intervals, (aPR = 0.49, 95%CI = 0.39–0.62 p = <0.001), mothers paying for IPTp drugs themselves (aPR = 0.74, 95%CI = 0.57–0.97, p = <0.031). The Details are shown in Tables 4 and 5.

## Discussion

Overall, a half of the mothers took optimal doses of IPTp with the least uptake recorded at Kyarumba PHC HCIII. This is far below the national target of 93% by 2020/2021 [13] but slightly higher than the 51% national IPTp+3 coverage for financial year 2020/21 as reported in the Annual Health sector performance report [4]. These results indicate that only 5 in every 10 mothers are believed to be having some protection from malaria and with a reduced risk of getting malaria related complications during pregnancy. Though the study in Malawi also reported 4 in every 10 mothers with optimal uptake of IPTp, they are all still below the WHO

**Table 4. Crude and adjusted analysis for demographic factors associated with uptake of IPTp among 396 mothers from 8 PNFP facilities in Kasese District.**

| Variable | Optimal n(%) (N = 204) | Sub optimal n(%) (N = 192) | Bivariate | | Multivariate | |
|---|---|---|---|---|---|---|
| | | | Crude PR 95% CI | p-value | Adjusted PR 95%CI | p-value |
| **Age in years** | | | | | | |
| Age in years ≤26 | 108(50.9) | 104(49.1) | | | | |
| >26 | 96(52.2) | 88 (47.8) | 1.02(0.84–1.24) | 0.808 | | |
| **Marital status** | | | | | | |
| Single | 38 (37.6) | 63(62.4) | Reference | | Reference | |
| Married | 153(56.9) | 116(42.1) | 1.51(1.15–1.99) | 0.003 | 1.35(1.06–1.7) | 0.014 |
| Divorced | 13(50.0) | 13(50.0) | 1.33(0.83–2.11) | 0.225 | 1.39(0.9–2.15) | 0.136 |
| **Level of education** | | | | | | |
| No education | 35(53.1) | 31(46.9) | Reference | | | |
| Primary | 99(49.5) | 101(50.5) | 0.93(0.75–1.22) | 0.612 | | |
| Post primary | 70(53.8) | 60(46.2) | 1.02(0.77–1.34) | 0.914 | | |
| **Residence** | | | | | | |
| Rural | 138(50.7) | 134(49.3) | Reference | | | |
| Urban | 66(53.2) | 58(46.8) | 1.05(0.86–1.29) | 0.645 | | |
| **Occupation** | | | | | | |
| Employed | 22(68.7) | 10(31.3) | Reference | | | |
| Self employed | 47(50.0) | 47(50.0) | 0.73(0.53–0.99) | 0.043 | | |
| Unemployed | 135(50.0) | 135(50.0) | 0.73(0.56–0.95) | 0.018 | | |
| **Distance from Health facility** | | | | | | |
| Less than 5Km | 127(55.2) | 103(44.8) | Reference | | | |
| More than 5Km | 77(46.4) | 89(53.6) | 0.84(0.68–1.02) | 0.087 | | |
| **Live with Husband** | | | | | | |
| Yes | 133(57.6) | 98(42.4) | Reference | | | |
| No | 20(52.6) | 18(47.4) | 0.01(0.66–1.26–0.94) | 0.584 | | |
| Not applicable | 51(40.2) | 76(59.8) | 0.01(0.52–0.64) | 0.003 | | |

PR: Prevelence ratios, CI: Confidence interval, Pv-Probability value

recommendation [14]. The moderately high coverage of IPTp in our study could have been due to the strict implementation of Result Based Financing (RBF) strategy at health facilities where optimal uptake of IPTp was among the key indicators of focus.

Additionally, interviewing mothers who had delivered within 72 hours might have led to this high results because this method reduced recall bias since some of them had attended ANC before, so it was easy for them to remember the drugs they had taken. These results were consistent with a cross-sectional study in private health facilities in Tema metropolis in Ghana with relatively same sample size that reported 46.6% optimal uptake among mothers who had delivered [15]. An earlier and related study from the public health facilities in eastern Uganda reported low optimum uptake of IPTp-SP and this was reportedly due to missed opportunities

Furthermore, the analysis of data from the Uganda Demographic and Health Survey (2016) on uptake of IPTp revealed coverage of 18% optimal doses of IPTp [16] lower than 41% reported in the Uganda malaria indicator survey [17]. Besides, this may indicate geographical variations in the intervention uptake across the country. Different study methodologies could also explain the difference. For example, the survey sampled respondents from the entire country while this study drew its sample from only one district.

From this study, married mothers had 35% higher chances of completing optimal doses. This may be explained by the emotional and financial support from their husbands as well as

**Table 5. Crude and adjusted analysis for behavioral, reproductive and knowledge factors associated with uptake of IPTp among 396 mothers from 8 PNFP facilities in Kasese District.**

| Variable | Optimal n(%) (N = 204) | Sub optimal n(%) (N = 192) | Bivariate Crude PR 95% CI | p-value | Multivariate Adjusted PR 95%CI | p-value |
|---|---|---|---|---|---|---|
| **ANC Visits** | | | | | | |
| Less than 4 | 90(42.6) | 121(57.4) | Ref | | Reference | |
| More than 4 | 114(61.6) | 71(38.4) | 1.45(1.19-1.75) | <0.001 | 1.29(1.09-1.54) | <0.001 |
| **Mothers escorted by their husbands for ANC** | | | | | | |
| Yes | 82(56.2) | 64(43.8) | Ref | | | |
| No | 122(48.8) | 128(51.2) | 0.86(0.71-1.05) | 0.152 | | |
| **Amount paid during ANC visit** | | | | | | |
| <$2.6 | 147(52.3) | 134(47.7) | Ref | | | |
| $2.96-$5.34 | 34(49.3) | 35(50.7) | 0.94(0.72-1.23) | 0.658 | | |
| >$5.34 | 23(50.0) | 23(50.0) | 0.96(0.70-1.31) | 0.775 | | |
| **Taking IPT as DOT(N=355)** | | | | | | |
| Yes | 171(59.2) | 118(40.8) | Ref | | | |
| No | 27(40.9) | 39(59.1) | 0.69(0.57-0.94) | 0.019 | | |
| **Taking IPTp at recommended time intervals (N=355)** | | | | | | |
| Yes | 145(71.4) | 58(28.6) | Ref | | Reference | |
| No | 53(34.9) | 99(65.1) | 0.49(0.38-0.62) | <0.001 | 0.49(0.39-0.62) | <0.001 |
| **Meeting the costs for IPTp drugs(N=355)** | | | | | | |
| Did not buy drugs | 150(60.2) | 99(39.8) | Ref | | Reference | |
| Self | 26(46.4) | 30(53.6) | 0.77(0.57-1.04) | 0.09 | 0.74(0.57-0.97) | 0.031 |
| Husband/Parent | 22(44.0) | 28(56.0) | 0.73(0.52-1.02) | 0.06 | 0.79(0.58-1.06) | 0.11 |
| **HIV Status** | | | | | | |
| HIV Positive | 16(51.6) | 15(48.4) | Ref | | | |
| HIV Negative | 188(51.5) | 177(48.5) | 0.99(0.69-1.42) | 0.991 | | |
| **Malaria status of mothers in last 3 months** | | | | | | |
| Yes | 45(43.3) | 59(56.7) | Ref | | | |
| No | 159(54.4) | 133(45.6) | 1.26(0.99-1.61) | 0.065 | | |
| **Parity** | | | | | | |
| Median(IQR) 2(1-4) | | | 0.99(0.95-1.05) | 0.842 | | |
| **Gravidity** | | | | | | |
| Prime-parous | 48(53.3) | 42(46.7) | Reference | | | |
| Multi-porous | 156(51.0) | 150(49.0) | 0.88(0.71-1.09) | 0.25 | | |
| **Gestation age at first Visit (weeks)** | | | | | | |
| Less than 12 weeks | 37(53.6) | 32(46.4) | Reference | | | |
| 12 or more weeks | 167(51.1) | 160(48.9) | 0.12(0.75-1.22) | 0.69 | | |
| **Mother's age at first pregnancy** | | | | | | |
| Median(IQR) 18(16-22) | | | 0.99(0.97-1.00) | 0.238 | | |
| **Mothers' Knowledge on IPTp** | | | | | | |
| Not adequate | 139(49.5) | 142(50.5) | Reference | | | |
| Adequate | 65(56.5) | 50(43.5) | 1.14(0.93-1.39) | 0.19 | | |

PR: Prevalence ratios, CI: Confidence interval, Pv-Probability value

reminders to keep appointments [18]. It could also be that single or divorced women get traumatized and stigmatized for getting pregnant out of matrimony and end up not taking IPTp hence a need for their social protection. Results from the six districts of Tanzania has also indicated married women as more likely to take optimal doses of IPTp compared to the single women [19]. Some studies have shown a positive association between presence of husband-wife discussion about health matters, such as family planning and utilization of various health services [20].

Contrary to our study, a cross sectional study of 239 respondents in Northern Nigeria reported mother's advanced age positively influencing optimal uptake of IPTp [21]. This could be explained by the fact that old mothers are expected to have vast experiences and knowledge on pregnancy related issues, this is likely to increase the uptake. Nonetheless, adolescent pregnant girls are less likely to attend an ANC and seek timely healthcare due to stigmatization and financial constraints.

Although level of education was not significantly associated with optimal uptake of IPTp in this study, Having secondary or higher level of education was associated with optimal uptake of IPTp among mothers in a study of prevention of malaria among women of reproductive age in Mozambique [22]. However, significant association was reported in Arusha among a slightly higher sample size of post-delivery mothers. [23]. In this study, the variations could be explained by the low education levels of the mothers at 50.2% and 16.7% for primary and no education among mothers respectively.

Our study did not find a significant association of occupation with optimal uptake of IPTp. However, in Ogbomoso, Oyo State, Nigeria, study results revealed an association of being employed with uptake of Optimal IPTp contrary to our study finding [24]. This difference could be due to equal access or in access to SP across all categories occupation e.g, knowledge level and or acceptability of the therapy. Thus, the larger sample in the Nigerian study was likely drawn from across wider geographic area and may likely contained significant variations in participant characteristics compared to the sample in the current study from few shared facilities in one district.

The mothers who live with their husbands were more likely to take optimal doses of IPTp as reported among postpartum women in Tandahimba district, Tanzania [25], However, in our study this had no significant association on the uptake of IPTp.

Additionally, we found that number of ANC visits was significantly associated with optimal uptake of IPTp dose where mothers who attended $\geq$4 visits were 29% more likely to take optimal IPTp versus mothers who attended less than 4 visits. Hence the need for motivation strategies for mothers to attend more ANC visits. This finding is consistent with reports among mothers at a university Hospital, Kumasi-Ghana in 2018 which indicated that the number of ANC visits positively influencing the recommended IPTp uptake [26]. These results are consistent with the aOR = 1.55 (95% CI = 1.34–1.80) reported in a similar cross sectional study in Malawi though they interviewed mothers with live birth in the 2 years preceding the survey of 2015 to 2016 Malawi Demographic and Health Survey [14]. Nonetheless the high ANC visits does not translate to optimal uptake of IPTp [27].

From this study, Mothers who did not take IPTp at the recommended intervals had 51% chances less likely to complete optimal doses of IPTp. The irregular attendances make mothers skip appointments hence fail to complete their doses. System and social challenges like lack of transport might have limited the mothers from keeping appointments as well as stock outs hence failing to adhere to the stipulated time intervals for taking drugs [28].

Compared with other studies, this study further demonstrated a significant association between mothers who bought IPTp drugs for themselves and optimal IPTp uptake indicating 26% less chances of completing optimal doses of IPTp. This is consistent with a qualitative

study finding in the eastern and west Nile regions that reported mothers paying $0.28 for IPTp drugs at the health facility [29]. This significant association can be due to that fact sometimes the mothers fail to buy the full prescribed doses of IPTp and end up getting demotivated to complete the dose.

Payment of user fees to access antenatal care has been pointed out as a key factor associated with optimal utilization of ANC services in a five-country analysis of national service provision assessment surveys [30] though it was not found significant in our study. However, majority (71.0%) of the respondents spent an average of $ 3 on each visit for ANC services. This could have also limited uptake of the services. Additionally, a study in Mali, has recommended removal of user fees for ANC and child birth services to increase their uptake [31].

In our study, only 36.9% of the husbands escorted their wives to health facilities for ANC and this was not found to significantly influence uptake of IPTp as reported in Kisungu HCIII [32]. This might have been due to the low power in our study to detect the detect the significant difference.

As observed among postpartum women in Tanzania's Tandahimba district [25], mothers who live with their husbands were more likely to take the recommended doses of IPTp contrary to our findings.

Knowledge of mothers on IPTp in this study was not significantly associated with optimal uptake of IPTp contrary to the study in Cameroon and Mali respectively [33, 34]. The current study being conducted in only private not for profit health facilities might have contributed to this none coherence findings with other studies.

Like in the cross sectional study in western Kenya in 2020, parity was not found to influence uptake of optimal IPTp [18]. This is contrary to the results from a cross sectional study in Sierra Leone among 8526 that reported higher parity (>4) being associated with lower odds of taking IPTp-SP. [35].

In a rural to urban prospective cohort study in Mali [36]. a positive association was reported among prime gravida than in multi-gravid, contrary to our study. This could probably be due to the high proportion of prime-gravid mothers (56%) in their study compared 22.7% in the current study.

In this study, mothers living with their husbands was not significantly associated with optimal uptake of IPTp as it was reported among women in Tandahimba district of Tanzania [25].

This study has some limitation. Self-reported adherence to uptake of IPTp especially for mothers who bought IPTp drugs from private clinics and took them from home made it not possible to truly ascertain whether they took the recommended doses, this might have introduced reporting bias hence over estimation of the outcome. Since the study was targeting postpartum mothers, some might have not recalled the drugs they took during their ANC visits leading to bias, However, this was minimized by showing samples of SP or cotrimoxazole to them in order for them to relate their responses to the drug in question. No causal inferences may be drawn from these findings because of snapshot nature of cross-sectional studies.

## Conclusion and recommendation

The study has established that about a half of the pregnant mothers completed ≥3 doses of IPTp-SP or ≥ for HIV negative pregnant mothers and 95% adherence to cotrimoxazole for HIV Positive mothers in private not for profit health facilities in Kasese. However, this is substantially lower than the national expected target of 80%. Based on the research results, being married, attending more than 4 ANC visits, taking IPTp drugs at recommended intervals and mothers buying IPTp drugs by themselves were significantly associated with optimal uptake of IPTp. Given the location of the PNFP facilities in the underserved remote areas in the district,

we therefore recommend Ministry of health to fully implement the Public Private Partnership for Health policy guidelines and set uniformly subsidized and affordable costs for essential maternal services, DHO's office to keep track of the IPTp supplies and re-allocate drugs from over stocked to under stocked health facilities and orient the health workers on Goal oriented antenatal as well as health workers sensitizing the mothers on the prevention and control of malaria in pregnancy

## Supporting information

**S1 Data. Supporting information files.**
(DTA)

## Acknowledgments

We thank all the people who were involved in the synthesis of this research up to its completion, and grateful to Kasese district health officer for the administrative support, study participants for consenting to participate in this study, the health facility in charges and research assistants for their involvement in this study, as well as the staff and students in the Clinical Epidemiology Unit, Makerere University.

## Author Contributions

**Conceptualization:** Julius Mutoro, Justus Barageine Kafunjo, Freddy Eric-Kitutu, Joan Kalyango, Iving Mumbere.

**Data curation:** Julius Mutoro, Justus Barageine Kafunjo, Freddy Eric-Kitutu, Joan Kalyango, Iving Mumbere.

**Formal analysis:** Julius Mutoro, Freddy Eric-Kitutu, Joan Kalyango, Iving Mumbere.

**Funding acquisition:** Julius Mutoro.

**Investigation:** Julius Mutoro, Justus Barageine Kafunjo.

**Methodology:** Julius Mutoro, Justus Barageine Kafunjo, Freddy Eric-Kitutu, Joan Kalyango, Iving Mumbere.

**Project administration:** Julius Mutoro.

**Resources:** Julius Mutoro.

**Software:** Iving Mumbere.

**Supervision:** Julius Mutoro, Justus Barageine Kafunjo, Joan Kalyango, Nathan Nshakira.

**Validation:** Julius Mutoro, Joan Kalyango, Iving Mumbere, Nathan Nshakira.

**Visualization:** Julius Mutoro, Joan Kalyango, Nathan Nshakira.

**Writing – original draft:** Julius Mutoro, Joan Kalyango.

**Writing – review & editing:** Julius Mutoro.

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
