## [Decision Letter · Decision Letter 0]

25 Sep 2023

PGPH-D-23-01526

Title: Level of and factors associated with optimal uptake of intermittent preventive treatment for malaria in pregnancy at private-not-for-profit health facilities in Kasese district

Dear Dr. Mutoro,

Thank you for submitting your manuscript to PLOS Global Public Health. After careful consideration, we feel that it has merit but does not fully meet PLOS Global Public Health’s publication criteria as it currently stands. Therefore, we invite you to submit a revised version of the manuscript that addresses the points raised during the review process.

Agreeing with the two reviewers, this topic has a significant public health importance therefore is important to get published. However, there are certain changes necessary to this article before it can be published. 

Please ensure to follow the author's guidelines with regard to in-text referencing.The authors may consult a professional English writer to revise the entire manuscript.

We look forward to receiving your revised manuscript.

Kind regards,

Preeti Mahato, Ph.D.

Academic Editor

Journal Requirements:

2. In the online submission form, you indicated that "Data set and study materials used during the study are available from corresponding author on reasonable request". All PLOS journals now require all data underlying the findings described in their manuscript to be freely available to other researchers, either 1. In a public repository, 2. Within the manuscript itself, or 3. Uploaded as supplementary information.

Additional Editor Comments (if provided):

Reviewers' comments:

Reviewer's Responses to Questions

**Comments to the Author**

1. Does this manuscript meet PLOS Global Public Health’s publication criteria? Is the manuscript technically sound, and do the data support the conclusions? The manuscript must describe methodologically and ethically rigorous research with conclusions that are appropriately drawn based on the data presented.

Reviewer #1: Yes

Reviewer #2: Yes

2. Has the statistical analysis been performed appropriately and rigorously?

Reviewer #1: No

Reviewer #2: Yes

3. Have the authors made all data underlying the findings in their manuscript fully available (please refer to the Data Availability Statement at the start of the manuscript PDF file)?

Reviewer #1: No

Reviewer #2: Yes

4. Is the manuscript presented in an intelligible fashion and written in standard English?

Reviewer #1: Yes

Reviewer #2: Yes

5. Review Comments to the Author

Reviewer #1: Review Comments to the Authors

Reviewer #1: This article presents the findings of a thorough assessment of data from an adequate sample size collected from private health institutions on women's use of IPTp for malaria in pregnancy. The authors find an array of risk factors for both suboptimal and optimal antimalarial drug uptake. In general, the analysis is quite thorough, however, I made some recommendations to improve it.

I have a few major comments and minor comments on this article, the manuscript needs careful copyediting to remove minor grammatical mistakes. please see the major and minor comments below:

1. Please consider incorporating publications in the background section that report on the national coverage of IPTp in Uganda rather than just the optimal doses, particularly from research using the national survey dataset (UDHS). The dose disparities across the district will then be reported, providing context for why the Kasese district was chosen and strengthening the justification for this study.

2. For the Result section, please consider reporting the proportions or percentage in n (%) format, i.e., 269(67.9%), not 269/396(67.9%) for easy readability in the entire document.

For the regression results report in (cPR; 95% CI) or (aPR; 95% CI) i.e. (aPR: 0.74, 95% CI: 0.57-0.97), no need to add the p-values in the entire document.

3. For improved validity, it is important to report how you scored the mothers’ array of responses about their knowledge of malaria in pregnancy and achieved the categorization into adequate and not adequate.

4. Instead of the step-wise selection technique for building models, please consider using the four-step approach for modeling survey data as recommended by Heeringa et al., 2017 and Hosmer and Lemeshow 2013. This will allow you to consider a variety of exposure variables from both statistical approaches and epidemiological interest, regardless of the significance level. In the multivariate model, for example, omitting "age in years, level of education, distance from health, and some other variables of epidemiological importance" to the uptake of IPTp-SP simply because the level of significance is not a usual practice in modeling building.

5. If it is possible, please consider a sub-group analysis for women who were escorted by husbands to the clinic compared to those who were not escorted by husbands. This will provide relevant insights for understanding the influence of spouse support on the optimal uptake of IPTp among pregnant.

6. In your discussion, please consider highlighting similarities and differences in the level of optimal uptake of IPTp reported in this in relation to other districts and other countries in East Africa.

Minor comments

Line 24, Poisson instead of poison

Line 31, please consider paraphrasing your recommendation about efforts to intensify marital imply in the context of the study as opposed to merely stating marital status

Line 46, please consider reviewing the WHO updated IPTp guideline for better interpretation, I think uptake of IPTp doses should be scheduled for every ANC visit for HIV-negative women not necessarily daily

Line 426, if it is possible, please consider de-identifying the dataset and upload in a publicly available platform, I would suggest you review the data availability statement if this response is sufficient for the journal.

Line 438 – 450, please consider reviewing the format of reporting author’s contributions for the journal.

Line 459, please format the reference list according to journal requirement not necessarily italicised.

Reviewer #2: Manuscript Number: PGPH-D-23-01526

Thank you for the opportunity to review this manuscript, a cross-sectional study involving 396 post-partum mothers in the postnatal wards of 8 PNFP health facilities in Kasese district. The authors found that over half of the sample had received the recommended optimum number of IPTp-SP against Malaria in pregnancy, a coverage comparable primarily to what has been observed in other studies, even though still below the national and international target. Malaria, especially among pregnant women, remains a troubling public health issue, placing a high burden on the national budgets of endemic countries and negating international efforts. Thus, I commend the authors for adding to the evidence of poor adherence to the interventions against malaria, especially among vulnerable populations, such as pregnant women.

Nevertheless, there are some issues that authors need to address to move the manuscript forward.

Major observations

Line 68 - Methods: general comment:

It is not clear what variables/indicators were collected from the selected women. The authors need to provide a Table of the specific variables/data collected from the women and how these were measured. For example, how was the question 'number of SP doses taken' administered? Was it mainly self-reported by the respondents? Did the authors review any other ANC records aside from HIV status (if so, at what point in time during the data collection was that done)?

This information is significant to appreciate better the commendable uptake of SP recorded in this study. For instance, a monthly dose of SP is comprised of three tablets. So, the probability that some mothers misreported a single dose as 3 doses (by count of tablets) is very high, and this could have inflated the responses to the question (since it was mainly respondent reported), particularly since the respondents were shown the SP tablets.

Line 79-87: It is commendable that the authors provided the number of health facilities in the study area and how many they selected (8/15 PNFP). They did well by focusing on the private, not-for-profit health facilities, which often escape the radar of many researchers, likely because some of these facilities want to be left alone to run things their way due to their "private" status.'

That notwithstanding, it is essential to contextualize the selection procedure to clarify how selection bias was controlled. Thus, the authors should clarify the following questions:

1. Why did they select 8 of the PNFP facilities, not more or less, and how did they arrive at the decision to select that number?

2. As these facilities were sampled 'purposively', it is important to have a pictorial idea of where each selected facility is located to know how spread out or close up they are relative to one another and their coverage area. Thus, it would be advantageous for the authors to provide a district map highlighting the estimated (or better specific) locations of each of the selected facilities relative to the geographical layout of the district.

3. Roughly, how many postnatal women attended/accessed each of the selected facilities relative to the estimated number of expected pregnancies per annum in the district in the previous year and in the current year of this study? It may be helpful to provide a Table containing the names of the selected facilities, their respective ANC populations (for the previous and current year of study, respectively), and the exact number (proportion) of postnatal women they selected from each facility.

Line 152 -Table 1:

1. The categories of Marital status add up to >100%. Please revise.

2. The authors indicated that 269 (67.9%) of their sample had said they were "married". However, those living with their husbands are more (305) compared to those (269) married: "mothers living with their husbands" = 396 (305 yes+91 no). Thus, one wonders if the authors had different applicable definitions for "married" and "husbands". Logically, the question of 'mothers living with their husbands' should apply to only the 269 married women. Only "married women should be asked about their "husbands" and could be living with them ("husbands"), unless the definition of husband was applied differently to different women in this study (which will be problematic).

Therefore, the output of 305 (77.1%) yes for mothers living with their husbands looks erroneous, as it is greater than the proportion of "married" women (67.9%).

Thus, there appears to be a possibility of misclassification, and this could influence the analytical statistics (modelling association) performed.

Lines 156-160: The subsequent Table (I presume should be Table 2) indicates that '185(46.7) mothers made 4 or more 4 visits'.

Lines 161-162: Previously, the authors said, "approximately a half 211 (53.3) of the mothers attended more at least 4 visits". Now, they say "Overall, 211/396 (53.3%) attended ANC care for less than 4 visits...". The two statements contradict each other and should be clarified.

Since the total sample is less than 422, it is unclear how 211 could attend 4 or more visits and how another 211 could attend less than 4 visits.

Lines 164-168: How exactly the "Knowledge of mothers on Malaria in pregnancy..." was measured is unclear. What questions were used, and how were these obtained and measured?

How did the authors classify "Not adequate" versus "adequate" knowledge - what cut-off point or benchmark was used, and how was that derived?

Line 182: The authors titled this section as "level of optimal uptake of three or more doses of IPTp", but they say "...204 mothers took =>3doses of IPTp-SP or =95% adherence to cotrimoxazole giving an overall prevalence of 51.5% (46.6-56.4).". Whether the authors are referring to IPTp, cotrimoxazole, or both drugs concerning the 95% adherence is very confusing. By "overall prevalence", do the authors mean to say they combined the prevalence of SP and cotrimoxazole?

The authors are advised to present findings on the main objective of the study - uptake of IPTp-SP, instead of muddying it with other therapies.

Uptake of SP across SSA is generally low, even in settings where such essential ANC services are freely provided.

Considering the following: the authors indicated that,

1. Close to half of the participants in this study 'lived >5Km from their health facilities'.

2. All participants 'had to pay some amount out-of-pocket to access ANC services',

3. Over half (53.3%) made less than the required minimum of 4 ANC visits, and

4. About 40% did not receive SP as DOT.

Therefore, it sounds fascinating that over "95% mothers (likely they mean to say 52%) took =>3 or more doses of IPTp-SP". Compared to the available literature on IPTp-SP uptake, the over 95% uptake recorded in this study looks exciting inasmuch as it needs some clarifications:

1. The authors should clarify how the variable/question of 'number of SP doses taken so far' was administered/measured. Please refer to the comment in the methods section.

Minor observations

Line 5: Please check the journal's standard to be sure the title is required at this stage.

General comment on Abstract:

Conventionally, abbreviations are not commonly used in the abstracts of scientific papers. Please check the journal's standard to ensure this is an exception. Otherwise, it is best to provide the complete definitions instead, albeit the word limitation imposed by the journal - "not more than 300".

Abstract

Line 15: "sub optimal" is better written as a word - 'suboptimal' or hyphenated - 'sub-optimal' instead.

Line 17: The sentence appears to be missing an article - 'the' and the preposition - 'of'. Please check and consider revising as follows: '...the level of and factors associated with the uptake of IPTp...'

Line 17: Note that "not for profit" is better written as a hyphenated expression - 'not-for-profit'

Line 20: What do "HC IVs" and "HC IIIs" mean?

Line 22: The word "semi-Structured" does not need to be capitalized unless there is a particular context-wise significance; in that case, this should be clarified.

Line 25: Please check and indicate clearly if "(46.6-56.4)" is the confidence interval (CI) of the "51.5%" uptake, as I presume.

Lines 29-32 (abstract's Conclusion): I presume that the authors intend a structured abstract. If so, please adjust the conclusion to be a separate paragraph to comply with a structured abstract format.

Lines 34-36 (Keywords): The keywords appear as 'list of abbreviations'.

Also, use punctuation to distinguish each keyword from the others clearly.

Background

Line 38: The introductory sentence appears to be missing a full stop. This should be fixed to break the sentence after the first citation "1".

Lines 30-40: Please consider revising as follows - 'Sub-Saharan Africa continuously has the highest incidence of malaria among the risky populations, translating to 95% of the global burden of malaria cases...'

Lines 42-43: Please provide the appropriate citation(s) for the information.

Since the information in the introductory paragraph is essential for the topic in question, the authors need to rearrange it to highlight the logical flow and clarity of thoughts. Thus, from global - continental - country/national - sub-national, etc. E.g., information in lines 44-45 could better come after lines 38-40. For instance: "Globally Malaria is a major public health problem with an estimated 219 million cases and 435,000 deaths [1]. Sub-Saharan Africa continuously has the highest incidence of malaria among the risky populations, translating to 95% of the global burden of malaria cases. In Africa, malaria in pregnancy poses a 3-4 times risk of miscarriage, low birth weight and increase stillbirth[5].

In Uganda, malaria remains the highest contributor (29.1%) of outpatient cases, with 10.9 % of all deaths [41].... Placental transmission among pregnant women accounts for about 45% of malaria transmissions in Uganda's high malaria transmission settings... In response, Uganda adopted...'

Lines 45-47: Please consider revising as follows - 'In response, Uganda adopted the WHO recommendations [6] which call for daily cotrimoxazole for pregnant women who are HIV positive, and at least 3 doses of IPTp-SP throughout for pregnant women who are HIV negative, schedule at one dose a month, beginning from the second trimester of pregnancy.'

Please move the citation to the end of the sentence since the remaining information about IPTp-SP is referenced from the same source.

Lines 47-50: Please consider revising as follows:

'Although an analytical cross sectional study linked optimal uptake of IPTp to 73.3% reduction in malaria infections and anemia by more than half [7], optimal uptake of IPTp remains very low (14.7%) across Uganda [8] due to multiple challenges.'

Lines 47-48: Please delete "...in Mbale Regional Referral Hospital among 184 mothers in labour..."

Line 49: Please remove the full stop between "...half" and the citation "[7]", since it appears that the sentence is continuous after the citation.

Line 52: Replace "... they still have gaps in ..." with 'there is still a challenge...'

Line 54: The "D" in "Directly observed treatment" must not be capitalized. Alternatively, it is better to capitalize each word or none to ensure uniformity.

Line 55: To use an abbreviation for the first time without its full meaning, it should have been previously defined. E.g., private not-for-profit (PNFP).

Note that abbreviations are generally not allowed in abstracts but in the main text by convention.

Line 57: Please revise " without its full meaning" as 'They charge for some services like...'.

Additionally, especially since the health facilities are not-for-profit, it is helpful to clarify if the charge for these services is captured by the health insurance - Uganda National Minimum Health Care Package or not. Is it charged to those without health insurance, or is it out of pocket even if one has the Uganda National Minimum Health Care Package?

Line 60: consider using 'hard-to-reach' instead of "hard to reach".

Line 84: Please end the sentence after "...was done” and resume the following sentence (beginning from "...these...") in its own merit.

Line 93: I assume the authors meant the square root of d (margin of error) instead of "d2" - "d2" is not the same as 'd-squared'.

Lines 100-101 and 103-105: It appears that the text in these lines has a different font size than the surrounding text. Please check (see the highlighted portion in the commented manuscript).

Lines 117-118: Provide appropriate citation(s) to support the 10%confounding criteria.

Lines 122-126: The sentence is too long and verbose. Please revise it into at least two sentences.

Line 133: Avoid using conjugations - "didn't" - in formal scientific writings.

Line 136: "un employed" should be written as a word, not separated. Please revise.

Line 152 -Table 1:

1. The categories of Marital status add up to >100%. Please revise.

Lines 156-160: Using punctuation to break the sentence into concise, readable, and clear formats would be more beneficial.

The authors may consult a professional English writer to revise the entire manuscript.

What the authors mean by "...majority 281 (71.0%) amount paid per ANC visit ..." is also unclear.

Line 156: Replace "approximately a half..." with 'A total of...'

Lines 162-163: Please clarify in what currency the payment of "...less than 10,000..." was made.

Lines 164-168: As suggested, please revise this and all long and winding sentences into simpler and clearer formats.

Line 165: It is unclear what the authors mean by "multi-porous" and if that could be applicable in the context of this study. Perhaps they want to say 'multi-parous' instead, as in multi-parity?

Line 178 - page 10: The Table numbers do not appear in a series. Please check and revise. Tables need to be clearly distinguished by different numbers.

“Table 1” (likely Table 2): “Who met costs for IPTp drugs(N=355)”. Please, replace "met" with 'paid' instead. Also, specify the currency.

Replace "gestation age" with 'gestational age' instead.

Line 182: Delete "...eligible..."

Line 189: Please double-check the presentation of all CIs. E.g., "CPR.=084, 95%CI 0.0.68-1.02" is problematic. I presume the authors meant to write 'CPR =0.84, 95%CI: 0.68-1.02'.

Line 193: "(... 0.1.61)"

Line 193: Replace "table 2 and 3" with 'tables 2 and 3'.

Table 4: Please check - the figures for ANC visits seem inconsistent throughout the manuscript. 'Table 2' indicates that about 46% made 4 or more ANC visits, while Table 4 shows 61.6%. By the way, 185 of 204 does not yield 61.6%. Perhaps I have misread the information presented.

Also note that while 121+71=192 for the sub-optimal, 90+185 is more than the optimal proportion of 204.

How was "Malaria status of mothers in last 3 months" measured - secondary medical record review or active laboratory investigations (RDTs + microscopy)?

How did the authors prevent or reduce the possibility of false-positive Malaria test results?

Table 4: Please see 'multi-parous' for "multi-porous".

See 'gestational age' for "gestation age".

Lines 215-376 - Discussion:

General observations

- the authors sub-headed the discussion, and this is unnecessary and unconventional. The format of a scientific manuscript should not be comparable to a compiled thesis. Please look at the Discussion sections of other published papers, albeit the author guide of the journal, and format accordingly.

Ideally, the recommendation paragraph should come before the paragraph on limitations.

- There appear to be inconsistencies in the in-text citations: in some cases, the authors use a full stop before the citation. In other cases, they use a comma, and sometimes, there is no punctuation. Please check and revise appropriately viz-a-viz other examples published in the journal.

- The discussion is generally too long and seemingly inflated by redundant information. The authors should consider revising it to be more concise.

Lines 216-219: Please delete lines 216-219, as the information does not add significance to the paper. A typical discussion should begin with a concise summary of the study's main findings. The authors can check other publications in the journal to serve as a guide.

Lines 221-227: The authors assert that 'about 6 of 10 women (60%) in the current study received the optimal number of SP doses, and they compared this finding to a Malawian study which reported 4 of 10 women optimal uptake.' This comparison is problematic for the following reasons:

1. 51.5% of optimum SP uptake is better translatable 5/10, not 6/10, because the difference is 10%, and 10% of 396 is approximately 40 more women.

2. The Malawian study reported a finding of 30.2%, which is 10% less than the authors quoted. So, one may argue that the authors made up their finding by +10% and the Malawian study by the same percentage. This adjustment (other than the reality) is not appropriate.

Consequently, the finding on optimal SP uptake in the current study is roughly 20% higher than reported in the Malawian study. Thus, the two studies are not necessarily comparable even though both are below the WHO and national targets. Therefore, it is better not to use the percentile scores as the basis of the comparison. Instead, please clarify that the basis of comparing the two studies is on the 'WHO benchmark of 93%'.

Lines 230-232: It is problematic to say that the moderate coverage of optimum uptake in this study is due to a subsidized payment system. No statistical evidence is presented in this study to support this claim.

The authors reported that "281(71.0)" - 7 out of 10 women - paid less than 10,000 (Shillings) to access the ANC services. There is no explicit guarantee that all the 51.9% of women who reportedly took 3 or more doses of SP were among the 71% who paid less than 10,000 (Shillings). Since the authors have not presented statistical evidence (significant positive association) between the amount of money spent on ANC services and uptake of SP in this study, it may be better to speculate the possible reasons for the optimal uptake (amidst other factors), other than the claim of 'subsidized cost'.

Please refer to attached report for continuation of Reviewer's comments and recommendations.

6. PLOS authors have the option to publish the peer review history of their article (what does this mean?). If published, this will include your full peer review and any attached files.

**Do you want your identity to be public for this peer review?** For information about this choice, including consent withdrawal, please see our Privacy Policy.

Reviewer #1: No

Reviewer #2: **Yes: **Dr. Frederick Dun-Dery

---

## [Author Response · Author response to Decision Letter 0]

3 Dec 2023

Reviewer 1: I have incorporated all your suggestions in the manuscript. Thanks for the corrections and your guidance 

Reviewer 2: I have incorporated all your suggestions in the manuscript. Thanks for your guidance and the corrections.

---

## [Editor Report · Decision Letter 1]

19 Dec 2023

PGPH-D-23-01526R1

Title: Level of and factors associated with optimal uptake of intermittent preventive treatment for malaria in pregnancy at private-not-for-profit health facilities in Kasese district

Dear Dr. Mutoro,

Thank you for submitting your manuscript to PLOS Global Public Health. After careful consideration, we feel that it has merit but does not fully meet PLOS Global Public Health’s publication criteria as it currently stands. Therefore, we invite you to submit a revised version of the manuscript that addresses the points raised during the review process.

We look forward to receiving your revised manuscript.

Kind regards,

Tinsae Alemayehu, MD

Guest Editor
---

## [Author Response · Author response to Decision Letter 1]

21 Feb 2024

I have responded to all the comments that were sent to me on different parts of the manuscript. I will wait for more guidance from you. Thanks for your support in ensuring that my work emerges as quality work.

---

## [Editor Report · Decision Letter 2]

23 Feb 2024

Title: Level of and factors associated with optimal uptake of intermittent preventive treatment for malaria in pregnancy at private-not-for-profit health facilities in Kasese district

PGPH-D-23-01526R2

Dear Mutoro,

We are pleased to inform you that your manuscript 'Title: Level of and factors associated with optimal uptake of intermittent preventive treatment for malaria in pregnancy at private-not-for-profit health facilities in Kasese district' has been provisionally accepted for publication in PLOS Global Public Health.

Best regards,

Tinsae Alemayehu, MD

Guest Editor